# Special transition and extraordinary phase on the surface of a two-dimensional quantum Heisenberg antiferromagnet

Chengxiang Ding[1⋆], Wenjing Zhu[2], Wenan Guo[2†] and Long Zhang[3‡]

**1** School of Microelectronics & Data Science, Anhui University of Technology, Maanshan, Anhui 243002, China
**2** Department of Physics, Beijing Normal University, Beijing 100875, China
**3** Kavli Institute for Theoretical Sciences and CAS Center for Excellence in Topological Quantum Computation, University of Chinese Academy of Sciences, Beijing 100190, China

⋆ dingcx@ahut.edu.cn , † waguo@bnu.edu.cn ,
‡ longzhang@ucas.ac.cn

## Abstract

Continuous phase transitions exhibit richer critical phenomena on the surface than in the bulk, because distinct surface universality classes can be realized at the same bulk critical point by tuning the surface interactions. The exploration of surface critical behavior provides a window looking into higher-dimensional boundary conformal field theories. In this work, we study the surface critical behavior of a two-dimensional (2D) quantum critical Heisenberg model by tuning the surface coupling strength, and discover a direct special transition on the surface from the ordinary phase into an extraordinary phase. The extraordinary phase has a long-range antiferromagnetic order on the surface, in sharp contrast to the logarithmic decaying spin correlations in the 3D classical O(3) model. The special transition point has a new set of critical exponents, $y_s = 0.86(4)$ and $\eta_{\parallel} = -0.33(1)$, which are distinct from the special transition of the classical O(3) model and indicate a new surface universality class of the 3D O(3) Wilson-Fisher theory.



# 1 Introduction

Exotic states of matter and unconventional phase transitions are the central topics of condensed matter and statistical physics. As the system undergoes a continuous phase transition in the bulk, its boundary also exhibits critical behavior [1, 2]. The surface critical behavior falls into different universality classes, which are controlled by the surface interactions and are richer than the corresponding bulk criticality. They correspond to the fixed points of the renormalization group (RG) transformation of the surface coupling parameters, and are captured by the conformally invariant boundary conditions of the boundary conformal field theory (BCFT) [3–5]. Therefore, the investigation of possible surface critical universality provides a window looking into the BCFT in complement to the bootstrap and holographic approaches [6–9].

The surface criticality has attracted renewed interest recently, which was partly motivated by the study of the gapless edge states of topological phases. A new surface universality class of the (2+1)-dimensional O(3) Wilson-Fisher quantum critical point (QCP) was observed in quantum antiferromagnetic (AF) Heisenberg models with gapless edge states composed of dangling spin-1/2 sites [10–13], but the necessity of the edge states was also questioned by the observation of similar surface critical exponents in spin-1 models [14], which do not have gapless edge states.

The surface critical behavior of the classical O($N$) models was believed to be well-documented in the literature [1,2,15,16]. In the three-dimensional (3D) Ising model ($N = 1$), the surface criticality is controlled by the ratio of the surface and the bulk coupling parameters, $\kappa = J_s/J$. For $\kappa$ smaller than a critical value $\kappa_c$, the surface criticality is only induced by the bulk phase transition at $T_{c,b}$ and belongs to the ordinary class. When $\kappa > \kappa_c$, the surface forms a long-range order below a critical temperature $T_{c,s} > T_{c,b}$. In this case, the surface is already ordered at $T_{c,b}$ but shows extra weak singularities, which is dubbed the extraordinary transition. At $\kappa = \kappa_c$, $T_{c,s}$ merges with $T_{c,b}$, leading to a special transition on the surface. These surface universality classes are characterized by a set of critical exponents.

The surface criticality of the 3D O($N$) models with $N \geq 2$ are more subtle, because the 2D surface itself cannot have long-range order for $T > T_{c,b}$ due to the celebrated Mermin-Wagner-Hohenberg theorem. In the 3D O(2) model, the surface undergoes a Berezinskii-Kosterlitz-Thouless transition at $T_{c,s} > T_{c,b}$ for $\kappa > \kappa_c$ and develops a quasi-long-range order for $T < T_{c,s}$, but the nature of the surface extraordinary phase at $T_{c,b}$ remained elusive [15]. In the 3D O(3) model, there is not any phase transition on the surface above $T_{c,b}$, thus it was not expected to show any special or extraordinary transitions, even though there was preliminary numerical evidence of a possible special transition in the strong coupling regime on the surface [15].

Motivated by the numerical evidence of a nonordinary transition in the (2+1)D O(3) models, a recent theoretical work [17, 18] proposed that the spin fluctuations on the surface is marginally irrelevant for $2 \leq N \leq N_c < \infty$ due to its coupling to the bulk critical state. The spin correlation on the surface is predicted to decay logarithmically, $C_{\parallel}(r) \propto [\log(r/r_0)]^{-q_{\parallel}}$. Therefore, such surface critical behavior is dubbed the extraordinary-log phase.

Guided by this proposal, the surface criticality of the 3D classical O(2) and O(3) models have been numerically revisited recently [19–21]. It is found that increasing the surface coupling strength leads to a special transition from the ordinary phase to an extraordinary phase in both models, and the spin correlation in the extraordinary phase decays logarithmically, which is consistent with the postulated extraordinary-log phase. The anomalous dimension of the magnetic order at the special transition of the O(3) model is numerically close to that in the (2+1)D quantum Heisenberg models with dangling spins [19].

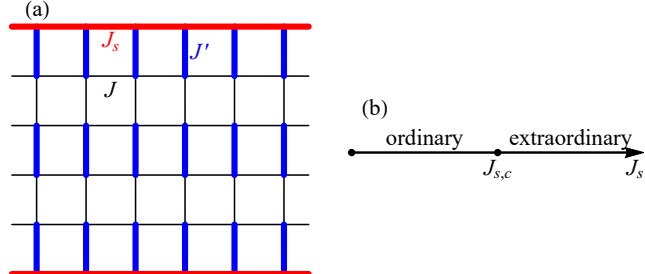

Figure 1: (a) The dimerized square lattice Heisenberg model with two open boundaries. $J'$ and $J$ are the coupling parameters of the strong (blue) and the weak (black) bonds in the bulk, repectively. $J_s$ is the coupling in the surface layer (red bonds). (b) Schematic phase diagram of the surface critical behavior at the bulk quantum critical point. A special transition is found at $J_{s,c} = 6.395(30)$ from the ordinary phase for $J_s < J_{s,c}$ to the extraordinary phase with a long-range antiferromagnetic order for $J_s > J_{s,c}$.

In a different theoretical approach [22], a spin-1/2 Heisenberg chain coupled to the ordinary surface of a (2+1)D O(3) QCP was studied with the non-Abelian bosonization and the RG analysis. It is shown that the Luttinger liquid phase of the spin chain is destabilized by its coupling to the bulk and gives the possibility of either an AF order or a valence bond solid (VBS) order, and there is a direct surface transition in between. The nonordinary surface critical state found in numerical simulations [10–13] was argued to correspond to the AF phase with a vanishingly small order parameter.

In this work, we reexamine the spin-1/2 quantum Heisenberg model on a dimerized square lattice, which is shown in Fig. 1 (a). At the bulk QCP, it was found to show the nonordinary critical behavior on the surface formed by cutting the strong bonds and exposing dangling spins, and show the ordinary critical behavior otherwise [11]. Starting from the ordinary phase of the nondangling surface, we show that increasing the surface coupling leads to a special transition into an extraordinary phase, which is illustrated in the surface phase diagram in Fig. 1 (b). The extraordinary phase has a long-range AF order on the surface, in sharp contrast to the extraordinary-log phase of the 3D classical O(2) and O(3) models. Moreover, we find that the critical exponents at the special transition are distinct from those at the special transition of the 3D classical O(3) model [19], thus indicate a new surface universality class of the 3D O(3) Wilson-Fisher theory.

## 2 Model and method

We study the spin-1/2 quantum Heisenberg model on the dimerized square lattice shown in Fig. 1 (a) with the open boundary condition in one direction and the periodic boundary condition in the other direction. The Hamiltonian is given by

$$H = J \sum_{\langle ij \rangle} \mathbf{S}_i \cdot \mathbf{S}_j + J' \sum_{\langle ij \rangle'} \mathbf{S}_i \cdot \mathbf{S}_j + J_s \sum_{\langle ij \rangle_s} \mathbf{S}_i \cdot \mathbf{S}_j, \tag{1}$$

where $J'$ and $J$ correspond to the strong (blue) and the weak (black) bonds in the bulk, and $J_s$ is the coupling parameter in the surface layer (red bonds). We set $J = 1$ as the unit of energy. This model undergoes a quantum phase transition in the bulk at $J'_c = 1.90951(1)$, which belongs to the 3D O(3) universality class [23–25]. For $J_s \simeq 1$, the surface shows the

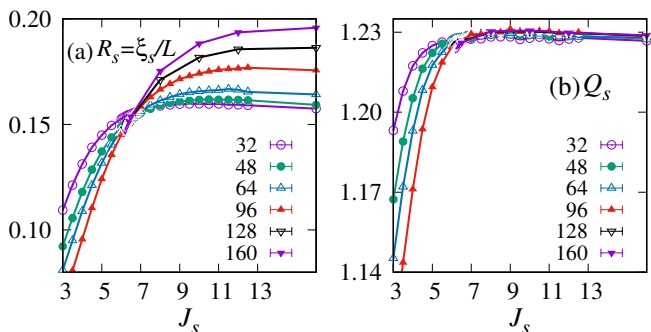

Figure 2: (a) The surface correlation ratio $R_s = \xi_s/L$ versus $J_s$ for different lattice sizes; (b) The Binder ratio $Q_s$ versus $J_s$ for different lattice sizes.

ordinary critical behavior [11]. On the other hand, for $J_s \gg 1$, the surface layer may be treated as a spin-1/2 Heisenberg chain with relatively weak coupling to the critical bulk state. We shall study the phase diagram of the surface critical behavior controlled by $J_s$ in this work.

We adopt the projective quantum Monte Carlo algorithm in the valence bond basis [26,27]. All simulations are performed at the bulk QCP. The largest system size in the simulations is $L = 160$. $10^7$ times of Monte Carlo sampling are taken for each data point.

The surface spin correlation function $C_\parallel(r)$, the surface-bulk spin correlation $C_\perp(r)$, the static spin structure factors $S(q)$, the surface correlation length $\xi_s$ and the Binder ratio $Q_s$ are calculated to characterize the surface critical behavior. Here,

$$C_\parallel(r) = \frac{(-1)^r}{L} \sum_x \langle \mathbf{S}_{(x,1)} \cdot \mathbf{S}_{(x+r,1)} \rangle, \, C_\perp(r) = \frac{(-1)^r}{L} \sum_x \langle \mathbf{S}_{(x,1)} \cdot \mathbf{S}_{(x,1+r)} \rangle. \tag{2}$$

The surface spin structure factor is defined as

$$S(q) = \langle \tilde{S}(q) \rangle, \tag{3}$$

where

$$\tilde{S}(q) = \frac{1}{L} \sum_{x,x'} e^{iq(x-x')} \mathbf{S}_{(x,1)} \cdot \mathbf{S}_{(x',1)}, \tag{4}$$

in which $q = \pi$ or $\pi + \delta q$ ($\delta q = 2\pi/L$). The surface correlation length $\xi_s$ is defined by

$$\xi_s = \frac{1}{2\sin(\pi/L)} \sqrt{\frac{S(\pi)}{S(\pi + \delta q)} - 1}. \tag{5}$$

The Binder ratio $Q_s$ is defined by

$$Q_s = \frac{\langle \tilde{S}^2(\pi) \rangle}{\langle \tilde{S}(\pi) \rangle^2}. \tag{6}$$

The special transition, the ordinary phase, and the nature of the extraordinary phase are derived from the finite-size scaling analysis of these quantities, which will be presented in detail in the following sections.

## 3 Results

### 3.1 Special transition

We first show that increasing the surface coupling strength $J_s$ induces a special transition on the surface by examining the surface correlation ratio $R_s = \xi_s/L$ and the Binder ratio $Q_s$, which

**SciPost**

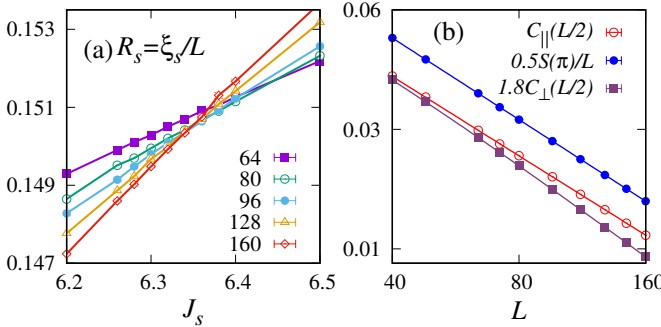

Figure 3: (a) The surface correlation ratio $R_s = \xi_s/L$ versus $J_s$ close to the special transition point $J_{s,c}$ for different lattice sizes. Solid lines are guide to the eye. (b) Log-log plot of $C_{\parallel}(L/2)$, $C_{\perp}(L/2)$ and $S(\pi)/L$ versus $L$ at the special transition $J_{s,c}$. Solid lines are fitting according to Eqs. (8), (9) and (10).

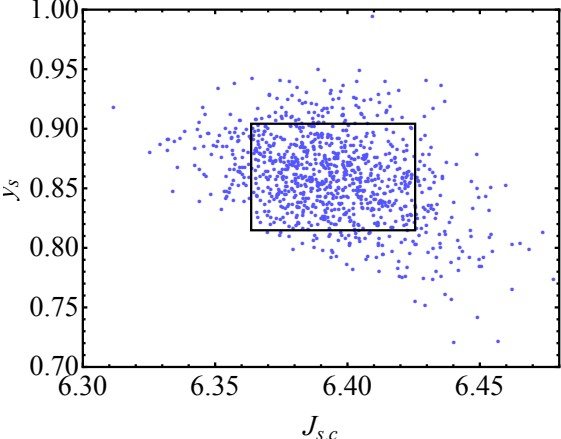

Figure 4: The scattered fitting parameters $J_{s,c}$ and $y_s$ extracted from 1000 sets of randomly generated $R_s$ data for $L_{\min} = 64$. The standard deviations (indicated by the black lines) serve as estimates of the statistical errors.

are plotted in Fig. 2. In the ordinary phase, $R_s$ decreases with increasing system size $L$, which is consistent with previous numerical results [10]. In contrast, $R_s$ is found to increase with $L$ in the large $J_s$ regime, which indicates stronger spin correlations on the surface. Therefore, the $R_s$ lines for different lattice sizes cross with each other, which indicates a critical point on the surface. The crossing of the Binder ratio lines also reveals such a special point. However, the data quality of the Binder ratio is not as good as $R_s$ in the vicinity of the critical point, and the correction to scaling is much stronger; hence we use the crossing of $R_s$ to locate the critical point. This is shown in Fig. 3 (a). If the transition is continuous with a scale-invariant critical point, the dimensionless ratio $R_s$ should take the following finite-size scaling form [28, 29],

$$R_s = \tilde{R}\big((J_s - J_{s,c})L^{y_s}\big) + \sum_i b_i L^{-\omega_i} = R_0 + \sum_{k=1}^{k_{\max}} a_k (J_s - J_{s,c})^k L^{k y_s} + \sum_i b_i L^{-\omega_i}, \qquad (7)$$

in which $y_s > 0$ is the scaling dimension of the relevant perturbation at the special transition, and $\omega_i$'s are the correction-to-scaling exponents. We set $\omega_1 = 0.759$ [30] and $\omega_2 = 2$ [31] in the following analysis. $\tilde{R}$ is the universal scaling function. In the second equality of Eq. (7), $\tilde{R}$ is expanded as a power series truncated at the $k_{\max}$-th order near the critical point. The critical point $J_{s,c}$ and the exponent $y_s$, together with the coefficients $R_0$, $a_k$'s, $b_1$ and $b_2$ are

fitting parameters in the data collapse analysis. The finite-size scaling correction is found to be quite strong, thus we gradually increase the smallest system size $L_{\min}$ in the analysis and achieve stable fitting for $L_{\min} \geq 64$. The results are listed in Table 1. The statistical errors of the fitting parameters are estimated with the following resampling method. A number of artificial data of $R_s$ are randomly generated from the original data and error bars assuming a normal distribution and fitted with the same data collapse scaling procedure. The extracted fitting parameters are scattered around those from the original data (see Fig. 4 for $L_{\min} = 64$), and the standard deviations are taken as the estimate of statistical errors of these fitting parameters. Our final estimates are $J_{s,c} = 6.395(30)$, $y_s = 0.86(4)$, and $R_0 = 0.154(1)$.

It should be noted that the finite-size scaling corrections arise from two sources, one is the leading correction proportional to $L^{-\omega_1}$, with $\omega_1 = 0.759$ for the current model, which comes from the irrelevant scaling field, another one is the background contribution analytic in $L^{-1}$. In practice, the analytic term $L^{-1}$ cannot be distinguished from $L^{-\omega_1}$ in the fitting procedure due to their close exponents. We also tried the fitting with $\omega_1 = 1$, the fitting quality is slightly worse and the difference of the results is very small, which falls in the range of the uncertainty of the error bars. Such a strategy has also been applied to all the other data fittings in this and the next subsections, although not explicitly stated.

The spin correlation functions $C_\parallel(L/2)$, $C_\perp(L/2)$, and the spin structure factor $S(\pi)/L$ at the special transition $J_{s,c}$ are shown in Fig. 3 (b). They are fitted according to the following finite-size scaling formulas,

$$C_\parallel(L/2) = L^{-1-\eta_\parallel}(a + bL^{-\omega}), \tag{8}$$

$$C_\perp(L/2) = L^{-1-\eta_\perp}(a + bL^{-\omega}), \tag{9}$$

$$S(\pi) = c + L^{2y_{h1}-3}(a + bL^{-\omega}), \tag{10}$$

in which $\eta_\parallel$, $\eta_\perp$ and $y_{h1}$ are the critical exponents. In (10), $c$ is the nonsingular part of $S(\pi)$, which comes from the contribution of short-range correlations. The fitting results, with $\omega = 0.759$ [30], are listed in Table 2. Our final estimates of the exponents are $\eta_\parallel = -0.33(1)$, $\eta_\perp = -0.18(1)$, $y_{h1} = 1.66(1)$. The estimates of $\eta_\parallel$ and $y_{h1}$ satisfy the scaling relation

$$\eta_\parallel = 3 - 2y_{h1}. \tag{11}$$

With the estimates of $\eta_\parallel$ and $\eta_\perp$, there is a slight deviation from the scaling relation

$$2\eta_\perp = \eta_\parallel + \eta, \tag{12}$$

where $\eta = 0.036$ is the bulk anomalous dimension [32]. This might be attributed to the inaccuracy of the estimated critical point $J_{s,c}$ and other systematic errors in the scaling analysis.

Table 1: Details of the finite-size scaling analysis of the surface correlation ratio $R_s = \xi_s/L$ in the vicinity of the special transition according to Eq. (7) with $k_{\max} = 2$. The correction-to-scaling exponents are set to be $\omega_1 = 0.759$ and $\omega_2 = 2$. The standard errors of the fitting parameters are obtained from the fitting procedure.

| $L_{\min}$ | $J_{s,c}$ | $y_s$ | $R_0$ | $\chi^2$/d.o.f |
|---|---|---|---|---|
| 48 | 6.387(10) | 0.859(24) | 0.1532(4) | 0.60 |
| 56 | 6.394(18) | 0.859(27) | 0.1535(7) | 0.63 |
| 64 | 6.395(19) | 0.858(27) | 0.1536(8) | 0.65 |

Table 2: Finite-size scaling analysis of $C_\parallel(L/2)$, $C_\perp(L/2)$, and $S(\pi)/L$ at the surface special transition according to Eqs. (8), (9), and (10) respectively.

| $C_\parallel(L/2)$ | $L_{\min}$ | $\eta_\parallel$ | $\chi^2/$d.o.f |
|---|---|---|---|
| | 48 | -0.340(4) | 1.27 |
| | 64 | -0.327(9) | 0.90 |
| $C_\perp(L/2)$ | $L_{\min}$ | $\eta_\perp$ | $\chi^2/$d.o.f |
| | 48 | -0.186(2) | 0.17 |
| | 64 | -0.184(2) | 0.17 |
| $S(\pi)$ | $L_{\min}$ | $y_{h1}$ | $\chi^2/$d.o.f |
| | 48 | 1.662(3) | 0.71 |
| | 64 | 1.658(3) | 0.46 |

In summary, the universality class of the special transition of the dimerized quantum Heisenberg model is described by the critical exponents

$$y_s = 0.86(4), \tag{13}$$
$$\eta_\parallel = -0.33(1), \tag{14}$$
$$\eta_\perp = -0.18(1), \tag{15}$$
$$y_{h1} = 1.66(1). \tag{16}$$

These exponents are drastically different from those obtained at the special transition of the 3D classical O(3) model [19], $y_s = 0.36(1)$ and $\eta_\parallel = -0.473(2)$, and previous numerical results with a dangling spin chain $\eta_\parallel \simeq -0.45$ [10, 11] and $\eta_\parallel \simeq -0.5$ [14]. Moreover, the critical exponents of $(4-\epsilon)$-dimensional O($n$) models at the special transition were calculated with the $\epsilon$-expansion [2, 33]:

$$\eta_\parallel = -\frac{n+2}{n+8}\epsilon + \frac{5(n+2)(4-n)}{2(n+8)^3}\epsilon^2 + O(\epsilon^3), \tag{17}$$
$$\phi_s = \frac{1}{2} - \frac{n+2}{4(n+8)}\epsilon + \frac{n+2}{8(n+8)^3}[8\pi^2(n+8) - (n^2 + 35n + 156)]\epsilon^2 + O(\epsilon^3). \tag{18}$$

Here, $\phi_s$ is the crossover exponent at the special transition, which is related to $y_s$ and the bulk correlation-length exponent $\nu$ by $y_s = \phi_s/\nu$. The $\epsilon$-expansion result of $\nu$ is given by [2, 34]

$$\nu = \frac{1}{2} + \frac{n+2}{4(n+8)}\epsilon + \frac{(n+2)(n^2 + 23n + 60)}{8(n+8)^3}\epsilon^2 + O(\epsilon^3). \tag{19}$$

Setting $n = 3$ and $\epsilon = 1$, we find $\eta_\parallel = -0.445$ and $y_s = 0.984$, and both are substantially different from our numerical results, which cannot be attributed to numerical errors. Therefore, the special transition found in this work belongs to a new surface universality class of the 3D O(3) Wilson-Fisher theory.

## 3.2 Ordinary phase

The surface is expected to be in the ordinary phase for $J_s < J_{s,c}$. The $J_s = 1$ case has been confirmed with the scaling behavior of $S(\pi)$, $C_\parallel$ and $C_\perp$ in Ref. [11]. Here, we focus on the $J_s = 2$ and $J_s = 3$ cases, the results of which are shown in Fig. 5.

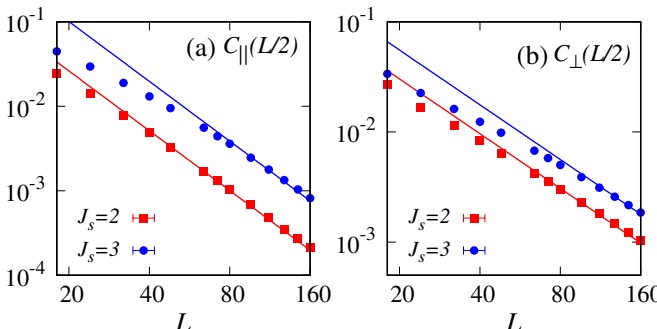

Figure 5: Log-log plot of the correlation functions in the ordinary phase: (a) $C_\parallel(L/2)$ versus $L$; (b) $C_\perp(L/2)$ versus $L$.

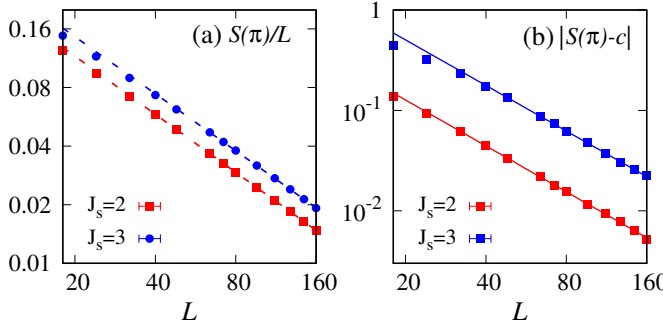

Figure 6: Scaling behavior of the spin structure factor $S(\pi)$ in the ordinary phase: (a) log-log plot of $S(\pi)/L$, where the dashed lines are proportional to $L^{-1}$; (b) log-log plot of the singular part of $S(\pi)$, i.e., $|S(\pi) - c|$.

The data of $C_\parallel(L/2)$ and $C_\perp(L/2)$ at $J_s = 2$ are fitted according to

$$C_\parallel(L/2) = L^{-1-\eta_\parallel}(a + bL^{-\omega}), \tag{20}$$

$$C_\perp(L/2) = L^{-1-\eta_\perp}(a + bL^{-\omega}), \tag{21}$$

with $\omega = 0.759$ [30]. This gives $\eta_\parallel = 1.35(2)$ and $\eta_\perp = 0.67(2)$. These results satisfy the scalig law (12) and are consistent with the ordinary phase of the 3D O(3) universality class. For the $J_s = 3$ case, the correction-to-scaling effect is much stronger as it is closer to the special transition; Nevertheless, we find the data of $C_\parallel(L/2)$ and $C_\perp(L/2)$ approach to be parallel to those at $J_s = 2$ in the log-log plot for large lattice sizes (see Fig. 5), which indicates the same critical exponents.

The scaling behavior of $S(\pi)/L$ is shown in Fig. 6 (a), which turns out to be dominated by the nonsingular constant $c$ in Eq. (10) contributed by the short-range correlations as the critical exponent $y_{h1} < 1.5$ in the case of ordinary transition. Fitting the data at $J_s = 2$ according to Eq. (10) with $\omega = 0.759$ [30], we find the critical exponent from the subleading singular term, $y_{h1} = 0.81(2)$, which is consistent with the ordinary transition as the $J_s = 1$ case in Ref. [11] and satisfies the scaling formula (11). This is shown in Fig. 6 (b), which also includes the $J_s = 3$ case. Therefore, we conclude that the surface critical behavior is consistent with the ordinary universality class for $J_s < J_{s,c}$.

## 3.3 Extraordinary phase

We then study the nature of the extraordinary phase for $J_s > J_{s,c}$. In light of the proposals that the surface spin correlation in the extraordinary phase may either decay logarithmically [17]

Table 3: Extraordinary-log fitting of $C_\parallel(L/2)$ and $S(\pi)/L$ at $J_s = 10$ and 16 according to Eqs. (22) and (23).

| $C_\parallel(L/2)$ | $L_{\min}$ | $q_\parallel$ | $r_0$ | $\chi^2/\text{d.o.f}$ |
|---|---|---|---|---|
| $J_s = 10$ | 48 | 1.80(6) | 0.57(8) | 1.62 |
| | 64 | 1.85(13) | 0.51(16) | 1.88 |
| | 72 | 1.85(20) | 0.51(26) | 2.26 |
| $J_s = 16$ | 48 | 0.824(23) | 3.64(25) | 3.04 |
| | 64 | 0.942(27) | 2.50(22) | 0.58 |
| | 72 | 0.95(5) | 2.4(4) | 0.68 |
| $S(\pi)/L$ | $L_{\min}$ | $q_\parallel$ | $L_0$ | $\chi^2/\text{d.o.f}$ |
| $J_s = 10$ | 48 | 4.09(9) | 0.074(11) | 3.46 |
| | 64 | 3.83(13) | 0.116(27) | 2.61 |
| | 72 | 3.73(19) | 0.14(5) | 2.83 |
| $J_s = 16$ | 48 | 2.63(4) | 0.77(6) | 4.82 |
| | 64 | 2.47(6) | 1.03(11) | 2.58 |
| | 72 | 2.36(5) | 1.27(13) | 1.27 |

or have long-range AF order [22], we examine both possibilities in the following analysis.

The data of $C_\parallel(L/2)$ and $S(\pi)/L$ at $J_s = 10$ and 16 are shown in Fig. 7, both of which are deep in the extraordinary phase. We first consider the extraordinary-log scaling form proposed in Ref. [17]. Suppose that the surface spin correlation decays logarithmically,

$$C_\parallel(r) \propto [\ln(r/r_0)]^{-q_\parallel} , \tag{22}$$

in which $r_0$ is a nonuniversal constant, then the structure factor $S(\pi)/L$ would decay logarithmically as a function of the lattice size $L$ with the same exponent $q_\parallel$,

$$S(\pi)/L \propto [\ln(L/L_0)]^{-q_\parallel} , \tag{23}$$

with another nonuniversal constant $L_0$. We find that $C_\parallel(L/2)$ and $S(\pi)/L$ can be fitted pretty well with the logarithmic form as shown in Figs. 7 (a) and (b) and Table 3. However, the extracted exponents $q_\parallel$ from $C_\parallel(L/2)$ and $S(\pi)/L$ are different from each other and vary significantly with $J_s$. We note that $J_s$ determines the bare value of the surface velocity $v_s$, and it has been shown with the RG analysis [17] that $v_s$ flows logarithmically slow towards the bulk velocity $v_b$ and can affect the apparent exponent $q_\parallel$ extracted from numerical results at finite length scales. While the variation of $q_\parallel$ with $J_s$ might be attributed to the disparity of the surface and the bulk velocities due to the above arguments, it does not explain the difference of $q_\parallel$ extracted from $C_\parallel$ and $S(\pi)$. Therefore, such inconsistency and non-universality indicate that the surface spin correlations cannot be captured by the extraordinary-log scaling.

We thus turn to the possibility of a true long-range AF order on the surface. Suppose $C_\parallel(r)$ can be captured by a polynomial of $1/r$ as $r \to \infty$,

$$C_\parallel(r) = m_s^2 + c_1 r^{-1} + c_2 r^{-2} + c_3 r^{-3} , \tag{24}$$

then $S(\pi)/L$ is given by

$$S(\pi)/L = m_s^2 + c_1' L^{-1} + c_1'' L^{-1} \ln L + c_2' L^{-2} , \tag{25}$$

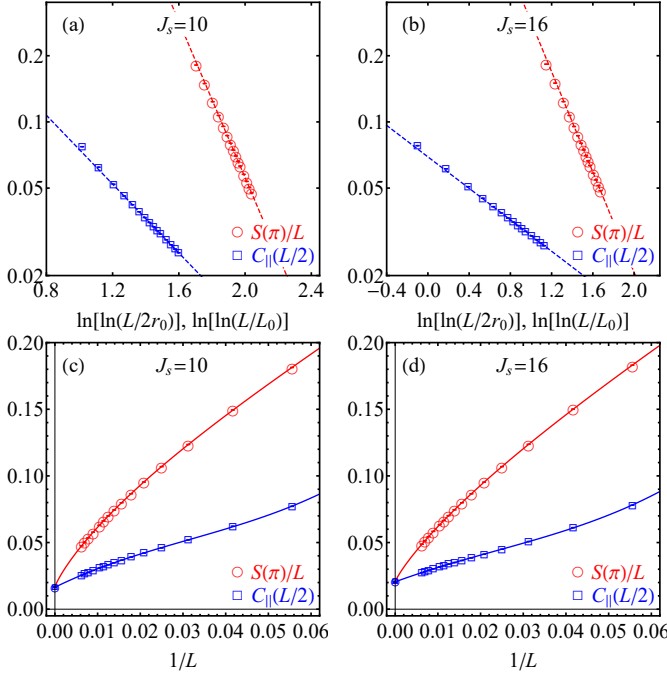

Figure 7: Finite-size scaling of $C_\parallel(L/2)$ and $S(\pi)/L$ at (a,c) $J_s = 10$ and (b,d) $J_s = 16$, both of which are deep in the extraordinary phase. (a,b) $C_\parallel(L/2)$ and $S(\pi)/L$ (in logarithmic scale) versus $\ln[\ln(L/2r_0)]$ and $\ln[\ln(L/L_0)]$, respectively. The nonuniversal constants $r_0$ and $L_0$ are presented in Table 3. Dashed lines are fitting with the logarithmic form in Eqs. (22) and (23). Their slopes indicate that the exponent $q_\parallel$ extracted from the two quantities are significantly different. (c,d) $C_\parallel(L/2)$ and $S(\pi)/L$ versus $1/L$. Lines are fitting according to Eqs. (24) and (25).

in which the $L^{-1}\ln L$ term comes from summing over the $r^{-1}$ term in $C_\parallel(r)$. The order parameter squared $m_s^2$ can be estimated by extrapolating $C_\parallel(L/2)$ and $S(\pi)/L$ to the thermodynamic limit, $m_s^2 = \lim_{L\to\infty} C_\parallel(L/2) = \lim_{L\to\infty} S(\pi)/L$. The orders of expansion in Eqs. (24) and (25) are restricted to keep the same number of fitting parameters. Fitting to the data of $C_\parallel(L/2)$ and $S(\pi)/L$, the results are shown in Figs. 7 (c) and (d). The two quantities yield consistent estimate of $m_s^2$ within one standard deviation, which justifies the above fitting procedure and indicates a long-range AF order.

More data for $6 \le J_s \le 16$ are presented in Fig. 8, and the extrapolated $m_s^2$ are shown in Fig. 9. The value of $m_s^2$ decreases with decreasing $J_s$ and becomes vanishingly small with large relative error bars near the special transition point $J_{s,c}$. A simple power-law fitting

$$m_s^2 \propto (J_s - J_{s,c})^{2\beta_\parallel} \tag{26}$$

gives an estimate of the critical point $J_{s,c} = 6.42(4)$, which is consistent with the previous estimate from the correlation ratio $R_s$. Therefore, we conclude that the surface has long-range AF order throughout the extraordinary phase $J_s > J_{s,c}$. In principle, $\beta_\parallel$ can also be extracted from fitting Eq. (26). However, our data of $m_s^2$ in the close vicinity of $J_{s,c}$ are not precise enough for a reliable estimate of $\beta_\parallel$. Instead, according to the scaling relation, $\beta_\parallel = (1 + \eta_\parallel)/2y_s$, we should have $\beta_\parallel = 0.40(3)$.

Furthermore, according to the definition of Eq. (4), in an ordered phase, $S(\pi + \delta q)$ should grow logarithmically (the constant term cancels out after summing, and the integral of the $1/r$ term contribute the logarithmic term), i.e., the data of $S(\pi + \delta q)$ should satisfy the finite-size

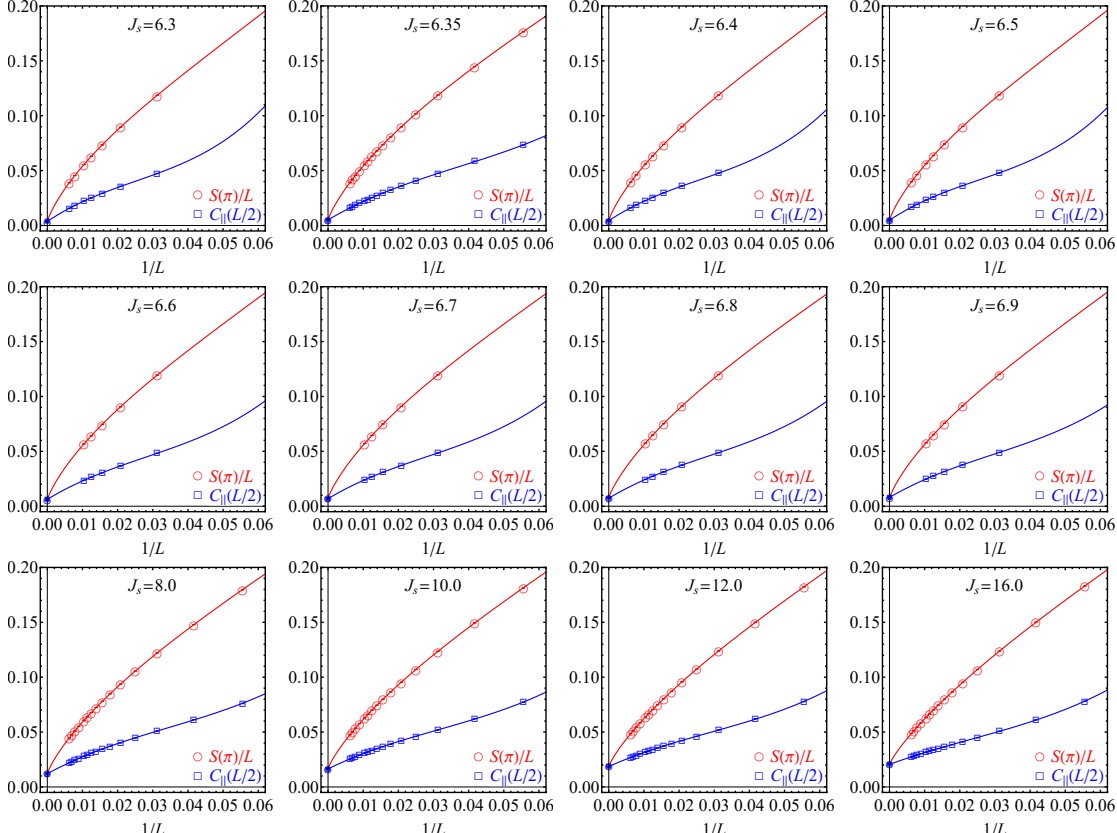

Figure 8: Finite-size scaling of $C_{\parallel}(L/2)$ and $S(\pi)/L$ for various surface coupling parameter $J_s$. Lines are fitting according to Eqs. (24) and (25).

scaling form

$$S(\pi + \delta q) = a + b \log(L), \tag{27}$$

combining with scaling of $S(\pi)$ in Eq. (25), we get the scaling formula of the square of the correlation ratio, which is written as

$$(\xi_s/L)^2 = a + bL/\log(L). \tag{28}$$

Figure 10 shows the scaling behaviors of $S(\pi + \delta q)$ and $(\xi_s/L)^2$ in the extraordinary phase, with $J_s = 16$, which further demonstrates that there is a long-range AF order.

## 4  Conclusion and discussions

In summary, we have found a special transition on the surface of a 2D quantum critical Heisenberg model between the ordinary phase and an extraordinary phase by tuning the coupling strength in the surface layer. The extraordinary phase has a long-range AF order, in sharp contrast to the extraordinary-log phase found in the 3D classical O(3) model. The critical exponents at the special transition are drastically different from those at the special transition of the classical O(3) model, thus indicate a new surface universality class of the 3D O(3) Wilson-Fisher theory.

The surface AF order observed in the extraordinary phase may be attributed to the long-range effective interactions induced by the critical fluctuations in the bulk, in the same spirit as the possible AF order proposed for a dangling spin chain coupled to the bulk in Ref. [22].

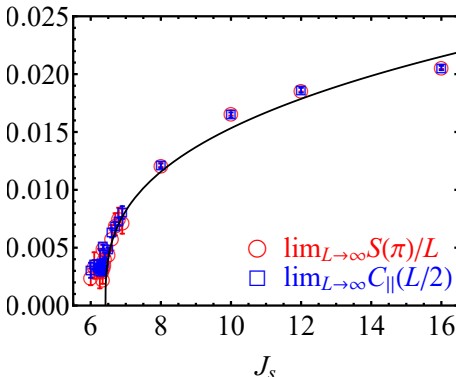

Figure 9: The extrapolated $m_s^2$ from $C_{\parallel}(L/2)$ (blue squares) and $S(\pi)/L$ (red circles) versus $J_s$. The solid line is the power-law fitting with Eq. (26).

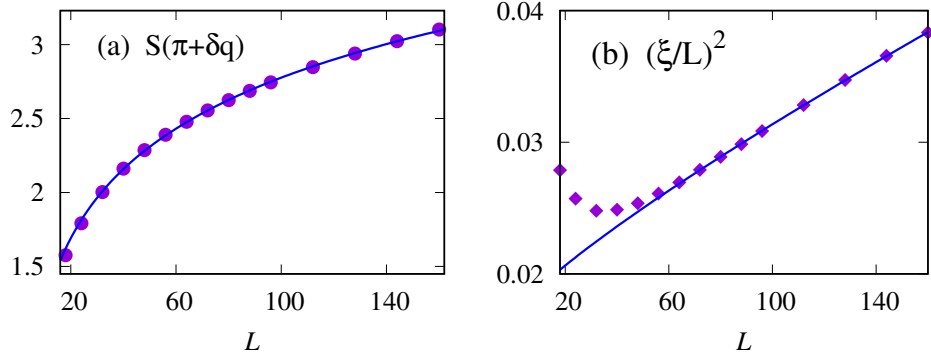

Figure 10: The scaling behaviors of (a) $S(\pi+\delta q)$ and (b) $(\xi_s/L)^2$ in the extraordinary phase, with $J_s = 16$; the solid lines are fitted to Eq. (27) and (28) respectively.

The AF order in the dangling-chain model is also confirmed numerically recently, which will be reported elsewhere.[1]

In Ref. [17], the extraordinary-log phase was proposed based on the perturbative RG analysis near the ordered fixed point at the 1D boundary. Starting from the ordered fixed point, spin fluctuations would lead to short-range correlations for a free-standing boundary, but the coupling with the bulk critical modes reverses the RG flow direction and makes the ordered fixed point stable. However, the logarithmically slow running towards this fixed point leads to the logarithmic decay of the spin correlation function instead of a long-range order, thus this is dubbed the extraordinary-log universality [17]. This has been confirmed in the 3D classical O(3) [19, 20] and O(2) [21] models, and the 3D AF three-state Potts model with emergent O(2) symmetry [35, 36]. However, the long-range AF order observed in the extraordinary phase of our current model and the distinct critical exponents at the special transition suggest that it might belong to a different regime of the 3D O(3) surface critical behavior, which is not captured by the perturbation from the normal surface fixed point. Instead, we may start from a possible gapless phase of the dangling ladder at the surface and treat its coupling to the bulk as perturbations following the similar method as Ref. [22]. However, the model studied in this work is different from the dangling-chain model. Here, the first two layers at the surface may be treated as a dangling ladder, which is weakly coupled to the bulk. The ladder in itself has a spin gap due to the interchain coupling, and the weak coupling to the bulk leads to the ordinary surface critical behavior. When the AF interaction in the surface layer is strong, we

---

[1]C. Ding and L. Zhang, in preparation.

may start from the Luttinger liquid phase of two decoupled chains and treat the interchain coupling and the coupling to the bulk as perturbations. In the whole phase space, the results of our numerical work may be far from the normal surface fixed point [17], hence different from the extraordinary-log behavior. With the bosonization and the RG analysis, we find that there is a phase with long-range AF order on the surface, and an ordinary-AF transition. The details will be presented elsewhere.[2]

## Acknowledgments

We thank Jing-Yuan Chen, Jian-Ping Lv, Francesco Parisen Toldin, Max A. Metlitski, Chao-Ming Jian, Cenke Xu, and Hong-Hao Song for helpful discussions and communications.

**Funding information** C.D. is supported by the National Science Foundation of China under Grants Number 11975024, the Anhui Provincial Supporting Program for Excellent Young Talents in Colleges and Universities under Grant Number gxyqZD2019023. L.Z. is supported by the National Key R&D Program (2018YFA0305800), the National Natural Science Foundation of China under Grants Numbers 12174387 and 11804337, CAS Strategic Priority Research Program (XDB28000000) and CAS Youth Innovation Promotion Association. W.Z. and W.G. are supported by the National Natural Science Foundation of China under Grants Numbers 12175015 and 11734002.

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
