# Peer review of "Special Transition and Extraordinary Phase on the Surface of a Two-Dimensional Quantum Heisenberg Antiferromagnet"

_SciPost Physics, doi:SciPost Phys. 15, 012 (2023)_

## Round 1 · Referee Report · Anonymous (Referee 1) · 2022-11-20

Report

Please see the attached pdf.

Attachment

  • validity: -
  • significance: -
  • originality: -
  • clarity: -
  • formatting: -
  • grammar: -

Author:  Chengxiang Ding  on 2023-03-03  [id 3427]

(in reply to Report 1 on 2022-11-20)

Reply to Referee 1: We have resubmitted our manuscript: https://www.scipost.org/submissions/scipost_202211_00001v2/; our answer to the questions are as follows:

  1. The authors study the boundary behavior of a 2d spin-1/2 model, whose bulk is tuned to a quantum critical point. They find a boundary phase transition as a function of boundary spin coupling between the ordinary boundary phase and an extraordinary boundary phase. They claim that the latter phase supports long-range antiferromagnetic order. Both the universality class of this transition and the nature of the extraordinary phase are claimed to be distinct from those in a classical 3D O(3) model. This is a surprising finding that runs counter to the theoretical arguments presented in Ref. 15. If correct, it will be of interest to large swaths of condensed matter, statistical mechanics and quantum field theory community, as it would be the first known example of spontaneous breaking of continuous symmetry on the 1d boundary of a 2d quantum critical state. However, I would invite the authors to present a more detailed analysis supporting their conclusions. In particular, I believe it will be useful for the reader to see.

    Reply: We thank the referee for appreciating the scientific merits of our research and the following useful comments and suggestions. The manuscript has been revised accordingly.

  2. A figure showing the correlation length (Eq. 4) as a function of system size in the extraordinary phase.

    Reply: In the revised manuscript, we plot the correlation ratio $R_s=\xi_s/L$ in a wide range of $J_s$ in Fig. 2 (a), which includes data deep in the extraordinary phase. The crossing of $R_s$ indicates a special transition.

  3. A figure showing the Binder ratio as a function of system size in the extraordinary phase.

    Reply: In the revised manuscript, we plot the Binder ratio $Q_s$ in a wide range of $J_s$ in Fig. 2 (b), which includes data deep in the extraordinary phase. The crossing of $Q_s$ also indicates a special transition.

  4. A figure showing the Binder ratio in the vicinity of the special transition, and an analysis of the location of the special transition and $y_s$ repeated using the Binder ratio.

    Reply: Thanks for the suggestion. Both the Binder ratio and the correlation ratio can in principle be used to locate the critical point, and the Binder ratio is indeed more widely used in the literature. However, we find that the data quality of the Binder ratio in the close vicinity of the special transition in this model is not as good as the correlation ratio, and its finite-size scaling correction seems much stronger, thus it is practically very difficult to locate an accurate special transition point with the Binder ratio. Therefore, we use the crossing of the correlation ratio $R_s$ to locate the critical point. This is clarified in the revised manuscript.

  5. An analysis of the exponent $\eta_\perp$ at the special transition - this can be used as a consistency check on the estimate of $\eta_\parallel$.

    Reply: Thanks for the suggestion. We have presented the surface-bulk correlation function $C_\perp(L/2)$ in the revised manuscript and performed the finite-size scaling analysis. The results are shown in Table 2. We find that the surface anomalous dimensions $\eta_\perp$ and $\eta_\parallel$ satisfy the scaling relation $2\eta_\perp=\eta_\parallel+\eta$, in which $\eta$ is the bulk anomalous dimension.

  6. An analysis of the boundary critical properties at some intermediate value of $J_s<J_{s,c}$ to confirm the ordinary nature of the boundary in this regime.

    Reply: We have performed simulations and finite-size scaling analysis at $J_s$=2 and 3. The results are presented in the newly added Section 3.2. The extracted critical exponents confirm the ordinary nature in this regime.

  7. Also, I would like to point out that the critical value $J_{s,c}\sim6.38$ reported by the authors is quite large and the simulations of the extraordinary phase are performed at even larger values of $J_s$. At such large values of $J_s$, ignoring the coupling of the boundary spin chain to the bulk, the boundary velocity is much larger than the bulk velocity. This velocity difference can potentially lead to slow cross-overs in the boundary behavior. In particular, such velocity anisotropies were considered in Ref. 15. It was shown that while in the extra-ordinary-log phase the surface velocity eventually flows to the bulk velocity, this flow is logarithmically slow and the exponent of the logarithmic fall-off $q_\parallel$ of the correlation function depends on the surface velocity. This could potentially explain the dependence of the exponent $q_\parallel$ reported by the authors in table 3 on $J_s$, although it does not immediately explain the difference of $q_\parallel$ extracted from $C_s$ and $S(\pi)$. Ideally, the authors would address the surface velocity by studying correlations along the time-direction. However, if their numerical algorithm does not easily allow for this, they should at least comment on the issue in the paper.

    Reply: We are grateful for these insightful comments, and have adapted them for the revised manuscript in Sec. 3.3: “However, the extracted exponents $q_\parallel$ from $C_s (L/2)$ and $S(\pi)/L$ are different from each other and vary significantly with $J_s$. We note that $J_s$ determines the bare value of the surface velocity $v_s$, and it has been shown with the RG analysis [Metlitski2022] that $v_s$ flows logarithmically slow towards the bulk velocity $v_b$ and can affect the apparent exponent $q_\parallel$ extracted from numerical results at finite length scales. While the variation of $q_s$ with $J_s$ might be attributed to the disparity of the surface and the bulk velocities due to the above arguments, it does not explain the difference of $q_\parallel$ extracted from $C_s$ and $S(\pi)$. Therefore, such inconsistency and non-universality indicate that the surface spin correlations cannot be captured by the extraordinary-log scaling.”

    While the bulk and surface velocities can in principle be extracted from the imaginary-time correlation functions, this is not easy for our projective quantum Monte Carlo algorithm. We thus leave this issue for future study.

---

## Round 1 · Referee Report · Aleix Gimenez-Grau (Referee 2) · 2022-12-6

Report

This work studies the critical behavior at the boundary of a quantum critical Heisenberg model. The authors start with a model with ordinary surface behavior, and by increasing the boundary couplings, they observe a transition into an extraordinary phase. Although it would be natural to identify this phase transition with the special transition, rather surprisingly their critical exponents are in tension with the classical O(3) model at the special transition. Another surprise is that the authors observe antiferromagnetic long-range order in the extraordinary phase, in contrast to the extraordinary-log phase recently proposed in ref. 15, and supported by Monte Carlo simulations of classical O(2) and O(3) models.

Since the results are in tension with theoretical expectations, they might lead to further effort by the community to understand the surface phase diagram for critical O(3) models. On one hand, it would be interesting if similar results were observed in simulations of different models. On the other hand, it would be good to have a theoretical explanation of the results in the present work.

For these reasons, I believe the results deserve publication, but it would be nice if some of the comments below can be addressed.

Requested changes

  • Althought refs. [6,7] might be appropriate for the holographic approach to BCFT, the boostrap papers 1210.4258 or 1502.07217 seem more relevant to the present discussion.
  • Below eq. (1) it is mentioned that previous work showed that for $J_s \simeq 1$ the surface has ordinary critical behavior. However, since $J_{s,c} = 6.38$ is larger, one might wonder if the ordinary phase is also observed for $1 < J_s < J_{s,c}$. Have the authors studied this and could maybe comment?
  • In fig. 2 the critical coupling is identified using the surface correlation length. Similar works, for instance ref. 17, use the surface Binder ratio. Did the authors try to use the surface Binder ratio? Are the two methods in good agreement, or is there a reason to prefer the correlation lenght, as it was done here?
  • The critical exponent $\eta_{||}$ is found from the parallel spin-spin correlation $C_{||}$. Could the authors also consider the boundary susceptibility $\chi_s$ and perpendicular spin-spin correlation $C_{\bot}$? This gives access to the exponents $y_{h_1}$ and $\eta_{\bot}$, which are related to $\eta_{||}$ by scaling relations. It would be valuable to see if these scaling relations are satisfied in the new special transition. For example, in ref 9 by three of the same authors, they observed the scaling relations were violated in some of their results. If these observables are hard to access in the present simulation, can the authors comment on that?
  • Below eqs. (9)-(10) one could also compare with the epsilon expansion. Interestingly, for the classical $O(3)$ model $y_s$ is quite far from the epsilon expansion, while the present results seem to be closer. Comparison to related works [8-10] might also be good.
  • Ref. [15] has a nice discussion of Monte Carlo results for boundary critical phenomena in quantum Heisenberg models. If the authors have an intuition on why their model violates the discussion in [15], it would be useful to comment on that.

If some of these are hard to address, it would be good if the authors could at least comment that in the text.

  • validity: -
  • significance: -
  • originality: -
  • clarity: -
  • formatting: -
  • grammar: -

Author:  Chengxiang Ding  on 2023-03-03  [id 3428]

(in reply to Report 2 by Aleix Gimenez-Grau on 2022-12-06)
Category:
answer to question

Reply to Referee 2(Aleix Gimenez-Grau): We have resubmitted our manuscript: https://www.scipost.org/submissions/scipost_202211_00001v2/;our answer to the questions are as follows:

This work studies the critical behavior at the boundary of a quantum critical Heisenberg model. The authors start with a model with ordinary surface behavior, and by increasing the boundary couplings, they observe a transition into an extraordinary phase. Although it would be natural to identify this phase transition with the special transition, rather surprisingly their critical exponents are in tension with the classical O(3) model at the special transition. Another surprise is that the authors observe antiferromagnetic long-range order in the extraordinary phase, in contrast to the extraordinary-log phase recently proposed in ref. 15, and supported by Monte Carlo simulations of classical O(2) and O(3) models.Since the results are in tension with theoretical expectations, they might lead to further effort by the community to understand the surface phase diagram for critical O(3) models. On one hand, it would be interesting if similar results were observed in simulations of different models. On the other hand, it would be good to have a theoretical explanation of the results in the present work. For these reasons, I believe the results deserve publication, but it would be nice if some of the comments below can be addressed.

$\mathbf{Reply}$: We thank Dr. Gimenez-Grau for appreciating the importance of our research and the following useful comments and suggestions. The manuscript has been revised accordingly.

  • Although refs. [6,7] might be appropriate for the holographic approach to BCFT, the bootstrap papers 1210.4258 or 1502.07217 seem more relevant to the present discussion.

$\mathbf{Reply}$: Thanks for the suggestion. These references are included in the revised manuscript (Refs. 8, 9).

  • Below eq. (1) it is mentioned that previous work showed that for $J_s\approx1$ the surface has ordinary critical behavior. However, since $J_{s,c}=6.38$ is larger, one might wonder if the ordinary phase is also observed for $1<J_s<J_{s,c}$. Have the authors studied this and could maybe comment?

$\mathbf{Reply}$: We have performed simulations and finite-size scaling analysis at $J_s=2$ and 3. The results are presented in the newly added Section 3.2. The extracted critical exponents confirm the ordinary nature in this regime.

  • In fig. 2 the critical coupling is identified using the surface correlation length. Similar works, for instance ref. 17, use the surface Binder ratio. Did the authors try to use the surface Binder ratio? Are the two methods in good agreement, or is there a reason to prefer the correlation length, as it was done here?

$\mathbf{Reply}$: Both the Binder ratio and the correlation ratio can in principle be used to locate the critical point, and the Binder ratio is indeed more widely used in the literature. The correlation ratio and the Binder ratio are plotted in a wide range of $J_s$ in Fig. 2 of the revised manuscript. The crossings of both quantities near $J_{s,c}$ indicate the special point. However, we find that the data quality of the Binder ratio in the close vicinity of the special transition in this model is not as good as the correlation ratio, and its finite-size scaling correction seems much stronger, thus it is practically very difficult to locate an accurate special transition point with the Binder ratio. Therefore, we use the crossing of the correlation ratio $R_s$ to locate the critical point. This is clarified in the revised manuscript.

  • The critical exponent $\eta_\parallel$ found from the parallel spin-spin correlation $C_\parallel$. Could the authors also consider the boundary susceptibility $\chi_s$ and perpendicular spin-spin correlation $C_\perp$? This gives access to the exponents $y_{h1}$ and $\eta_\perp$, which are related to $\eta_\parallel$ by scaling relations. It would be valuable to see if these scaling relations are satisfied in the new special transition. For example, in ref 9 by three of the same authors, they observed the scaling relations were violated in some of their results. If these observables are hard to access in the present simulation, can the authors comment on that?

$\mathbf{Reply}$: Thanks for the suggestion. In the revised manuscript, we have presented the staggered surface spin structure factor $S(\pi)$, which has the same scaling form as $\chi_s$, and the surface-bulk correlation function $C_\perp(L/2)$, and performed the finite-size scaling analysis. The results are shown in Table 2. We find that the critical exponents $y_{h1}$, $\eta_\perp$ and $\eta_\parallel$ atisfy the scaling relations at the new special transition point.

  • Below eqs. (9)-(10) one could also compare with the epsilon expansion. Interestingly, for the classical O(3) model $y_s$ is quite far from the epsilon expansion, while the present results seem to be closer. Comparison to related works [8-10] might also be good.

$\mathbf{Reply}$: Thanks for the suggestion. We have compared our results with those from the $\epsilon$-expansion of ($4-\epsilon$)-dimensional O(n) models at the special transition and previous numerical results in the literature, and find that there are significant differences, which cannot be attributed to numerical errors. The comparison is presented in the revised manuscript. Therefore, we conclude that our results indicate a new universality class of the surface critical behavior of the 3D O(3) model.

  • Ref. [15] has a nice discussion of Monte Carlo results for boundary critical phenomena in quantum Heisenberg models. If the authors have an intuition on why their model violates the discussion in [15], it would be useful to comment on that. If some of these are hard to address, it would be good if the authors could at least comment that in the text.

$\mathbf{Reply}$: Thanks for the suggestion. In the last section of the revised manuscript, we briefly recall the perturbative RG analysis near the normal fixed point by Metlitski and argue that the surface AF order might belong to a different regime of surface critical behavior. Instead, we suggest starting from a possible critical state of the dangling-ladder surface and treating its coupling to the bulk as perturbations, in the same fashion as the theoretical analysis of the dangling-chain model by Jian et al, which might explain the long-range AF order on the surface.

---

## Round 1 · Referee Report · Anonymous (Referee 3) · 2022-12-16

Strengths

-Timely topic
-Important advance

Weaknesses

-Technical analysis of MC data (see comments below)
-Presentation can be improved

Report

The authors use quantum Monte Carlo simulations to study the boundary critical behavior of a bidimensional quantum spin model.
While the bulk quantum phase transition belongs to the classical 3D Heisenberg universality class, its boundary behavior is less understood.
Contrary to some recent theoretical analysis, the authors find a boundary phase transition to an ordered phase.
Moreover, the critical exponents at special boundary transition separating the ordinary from the (boundary) ordered phase are significantly different than the classical ones.

Given the recent renewed interest in boundary critical behavior, the topic of the paper is certainly timely and appropriate to the journal.
Nevertheless, considering the fact that the results are unexpected, before considering the publication in Scipost Physics I ask the authors to answer the comments below.

Requested changes

Technical Remarks:

1.
Fits to Eq. (5) are done fixing omega=1, 2. The leading correction to scaling is due to the leading irrelevant bulk operator, which gives omega approximately 0.8. On the top of that, boundaries give rise to additional corrections with omega=1.
Since it would be very hard to distinguish two source of corrections with similar omega exponents, the authors should repeat the fits fixing omega = 0.759; this value is from PRB 102, 024406 (2020). A comparison with the fitted value with omega=1 should give a more reliable error bar.
On the other hand, the value omega=2 is unjustified.

2.
Even if the fits to Eq. (11)-(12) give an exponent which varies within the extraordinary phase, it is unclear how (for a given $J_s$) $q_\parallel$ is so different between fits to C(r) and fits to $S(\pi)$, since after all $S(\pi)$ is the integral of C(r). I would ask the author to double check the data and normalization.

3.
What is the justification of Ansatzes of Eq. (13) and (14)?

4.
A "smoking gun" of the extraordinary-log phase is the presence of logarithmic corrections to the Finite-size scaling behavior of various RG-invariants.
In particular, Refs. [17,18] show that the scaled spin stiffness $L\Upsilon$ and that $(\xi/L)^2$ are both $\propto \ln(L)$.
Considering also that the magnetization found in the extraordinary phase is rather small, in order to exclude the extraordinary-log phase I would ask the authors to provide a plot of $(\xi/L)^2$ and, if possible, of $L\Upsilon$ and the Binder ratio in the extraordinary phase.
In particular, if the extraordinary phase is ordered as reported in the paper, $(\xi/L)^2 \propto L$, while in the extraordinary-log phase $(\xi/L)^2\propto \ln(L)$.

Remarks on the presentation

5.
As discussed in Ref. [15], Sec. 4.1., in the extraordinary-log phase the surface velocity exhibits a logarithmic approach to the bulk velocity.
Considering that the simulations in the extraordinary phase reach rather large values of the coupling constants $J_s$, this slow approach can potentially influence the reliability of the extrapolated fit results. The authors should add a comment on that.

6.
The results of the paper are in contrast with the theoretical analysis of Ref. [15]. In the conclusion part, the manuscript should discuss this discrepancy, perhaps briefly recalling the arguments of Ref. [15].

  • validity: -
  • significance: -
  • originality: -
  • clarity: -
  • formatting: -
  • grammar: -

Author:  Chengxiang Ding  on 2023-03-03  [id 3430]

(in reply to Report 3 on 2022-12-16)

Reply to Referee 3:
We have resubmitted our manuscript: https://www.scipost.org/submissions/scipost_202211_00001v2/;
our answer to the questions are as follows:

The authors use quantum Monte Carlo simulations to study the boundary critical behavior of a bidimensional quantum spin model. While the bulk quantum phase transition belongs to the classical 3D Heisenberg universality class, its boundary behavior is less understood. Contrary to some recent theoretical analysis, the authors find a boundary phase transition to an ordered phase. Moreover, the critical exponents at special boundary transition separating the ordinary from the (boundary) ordered phase are significantly different than the classical ones.
Given the recent renewed interest in boundary critical behavior, the topic of the paper is certainly timely and appropriate to the journal. Nevertheless, considering the fact that the results are unexpected, before considering the publication in Scipost Physics I ask the authors to answer the comments below.

Technical Remarks:

*. Fits to Eq. (5) are done fixing $\omega$=1,2. The leading correction to scaling is due to the leading irrelevant bulk operator, which gives omega approximately 0.8. On the top of that, boundaries give rise to additional corrections with $\omega$=1. Since it would be very hard to distinguish two source of corrections with similar omega exponents, the authors should repeat the fits fixing $\omega$=0.759;
this value is from PRB 102, 024406 (2020). A comparison with the fitted value with $\omega$=1 should give a more reliable error bar.
On the other hand, the value $\omega$=2 is unjustified.

Reply: In the revised manuscript, we have carried out the data fitting with the leading irrelevant exponent $\omega_1=0.759$ and the next-to-leading exponent $\omega_2=2$, and found that the fitting quality turns out to be slightly better. The results are consistent with the previous results and are presented in Table I. The next-to-leading correction term is necessary; Otherwise, the value of $\chi^2/d.o.f$ is too large, indicating a low quality of fitting. For example, fitting without the $b_2L^{-\omega_2}$ term leads to $\chi^2/d.o.f=4.88$ for $L_{min}=48$.

At a generic critical point, finite-size scaling corrections arise from two sources: singular terms in $L^{-1}$ from irrelevant scaling operators, e.g., the leading correction to scaling proportional to $L^{-\omega_1}$ with $\omega_1=0.759$,
and background contributions analytic in $L^{-1}$. In practice, the first analytic term proportional to $L^{-1}$
cannot be distinguished from $L^{-\omega_1}$ in the fitting procedure due to their close exponents, thus only the second analytic term $L^{-2}$ is included as the subleading correction term, i.e., setting $\omega_2=2$. This practice was also adopted in PRB 30, 6615 (1984).
We thank the referee for drawing our attention to the reference PRB 02, 024406 (2020). This paper is cited in the revised manuscript.

*. Even if the fits to Eq. (11)-(12) give an exponent which varies within the extraordinary phase, it is unclear how (for a given $J_s$) $q_\parallel$ is so different between fits to C(r) and fits to $S(\pi)$, since after all $S(\pi)$ is the integral of C(r). I would ask the author to double check the data and normalization.

Reply: We have carefully double checked the data and normalization in the fitting. The different exponents of $q_\parallel$
from $C_\parallel$ and $S(\pi)$ imply the inconsistency in the ansatz of logarithmic decay in the extraordinary phase.
Therefore, we turn to the possibility of a long-range AF order.

*. What is the justification of Ansatzes of Eq. (13) and (14)?

Reply: Suppose that there is a long-range AF order on the surface, then the spin correlation function $C_s(r)$ can be derived
from the spin wave theory. This gives a long-range staggered correlation plus an analytic function in $1/r$,
which can be expanded in power series of $1/r$ as Eq. (13) [Eq. (25) in the revised manuscript].
Such a fitting form was adopted in, say, S. Sorella, Y. Otsuka, and S. Yunoki, Sci. Rep. 2, 992 (2012).
The form of the spin structure factor $S(\pi)/L$ as Eq. (14) [Eq. (26) in the revised manuscript] is obtained
from that of $C_s(r)$ by summing over r from 0 to L, in which the $L^{-1}\ln L$ term comes from the $r^{-1}$ term in $C_s(r)$.

*. A "smoking gun" of the extraordinary-log phase is the presence of logarithmic corrections to the Finite-size scaling
behavior of various RG-invariants. In particular, Refs. [17,18] show that the scaled spin stiffness $L\Upsilon$ and that $(\xi/L)^2$are
both $\propto \ln L$. Considering also that the magnetization found in the extraordinary phase is rather small,
in order to exclude the extraordinary-log phase I would ask the authors to provide a plot of $(\xi/L)^2$ and, if possible,
of $L\Upsilon$ and the Binder ratio in the extraordinary phase. In particular, if the extraordinary phase is ordered as reported in the paper,
$(\xi/L)^2\propto L$ while in the extraordinary-log phase $(\xi/L)^2\propto \ln L$.

Reply: We would thank the referee for this great suggestion. The data of $(\xi/L)^2$ and the Binder ratio in the extraordinary phase
are presented in Fig. 10 and Fig. 2 of the revised manuscript, respectively. It is a pity that we did not compute the spin stiffness.
As shown in Fig. 10, the $(\xi/L)^2$ in the extraordinary phase clearly follows the linear scaling as expected for a long-range AF order.

*. As discussed in Ref. [15], Sec. 4.1., in the extraordinary-log phase the surface velocity exhibits a logarithmic approach
to the bulk velocity. Considering that the simulations in the extraordinary phase reach rather large values of the
coupling constants $J_s$, this slow approach can potentially influence the reliability of the extrapolated fit results.
The authors should add a comment on that.

Reply: This issue was also raised by Referee 1. We have adapted his/her comments for the revised manuscript in Sec. 3.3:
"However, the extracted exponents $q_\parallel$ from $C_s(L/2)$ and $S(\pi)/L$ are different from each other and vary significantly with $J_s$.
We note that $J_s$ determines the bare value of the surface velocity $v_s$, and it has been shown with the RG analysis [Metlitski2022]
that $v_s$ flows logarithmically slow towards the bulk velocity $v_b$ and can affect the apparent exponent $q_\parallel$ extracted from
numerical results at finite length scales. While the variation of $q_\parallel$ with $J_s$ might be attributed to the disparity
of the surface and the bulk velocities due to the above arguments, it does not explain the difference of $q_\parallel$
extracted from $C_\parallel$ and $S(\pi)$. Therefore, such inconsistency and non-universality indicate that
the surface spin correlations cannot be captured by the extraordinary-log scaling."

While the bulk and surface velocities can in principle be extracted from the imaginary-time correlation functions, this is not easy for our projective quantum Monte Carlo algorithm. We thus leave this issue for future study.

*. The results of the paper are in contrast with the theoretical analysis of Ref. [15]. In the conclusion part,
the manuscript should discuss this discrepancy, perhaps briefly recalling the arguments of Ref. [15].

Reply: Thanks for the suggestion. In the last section of the revised manuscript, we briefly recall the perturbative RG analysis near the normal fixed point by Metlitski and argue that the surface AF order might belong to a different regime of surface critical behavior. Instead, we suggest starting from a possible critical state of the dangling-ladder surface and treating its coupling to the bulk as perturbations, in the same fashion as the theoretical analysis of the dangling-chain model by Jian et al, which might explain the long-range AF order on the surface.

---

## Round 2 · Referee Report · Aleix Gimenez-Grau · 2023-3-31

Report

The authors have addressed many of the points raised in the three referee reports, so I believe the paper is almost ready for publication.

However, in reading v2 I noticed the staggered magnetization Cs is no longer defined, but it still appears in some equations. Could the authors please fix this? There is also a typo "ordianry" below eq. (7). Finally, although the conclusion now discusses briefly the work by Metlitski, I strongly encourage the authors to provide a more thorough discussion, since this will definitely increase the quality of the paper.

  • validity: -
  • significance: -
  • originality: -
  • clarity: -
  • formatting: -
  • grammar: -

Author:  Chengxiang Ding  on 2023-05-02  [id 3637]

(in reply to Report 1 by Aleix Gimenez-Grau on 2023-03-31)

Thanks very much!

  1. We rewrite the definition of $C_\parallel$ as $C_{\parallel}(r)=\frac{(-1)^r}{L}\sum_{x}\langle \mbf{S}{(x,1)}\cdot\mbf{S}\rangle$ in Eq. (2).

    All the $C_s$ in the text have been changed to $C_\parallel$.

  2. The typo "ordianry" is fixed.

  3. In Ref. [17], the extraordinary-log phase was proposed based on the perturbative RG analysis near the normal fixed point at the 1D boundary. Starting from the normal fixed point, where the spins show an infinitesimal long-range order, the spin interactions would be relevant and lead to short-range correlations at a free-standing boundary, but the coupling with the bulk critical modes reverses the RG flow direction and makes the normal fixed point stable. However, the logarithmically slow running towards this fixed point leads to the logarithmic decay of the spin correlation function instead of a long-range order, thus this is dubbed the extraordinary-log universality.

In the whole phase space, the results of our numerical work may be far from the normal surface fixed point, hence different from the extraordinary-log behavior.

This discussion is added to section 4.

---

## Round 2 · Referee Report · Anonymous · 2023-4-6

Report

The authors have positively responded to my comments, thus I think the paper is suitable for publication.
Still, concerning the fits of Sec. 3.1 and 3.2 (my comment 1. in the previous report):

On the basis of RG, one expects corrections L^(-0.759) *and* corrections L^(-1). But of course, the two are practically impossible to be distinguished.
Therefore, one should, at minimum, try omega_1=0.759 *and* try omega_1=1.
If, as it is likely, the fits results are the same, this should be briefly indicated in the paper.
If there is a significant discrepancy between fits with omega_1=0.759 fits omega_1=1, then the final uncertainty should take such discrepancy into account.

I would encourage the authors to briefly include a short discussion along these lines.

  • validity: -
  • significance: -
  • originality: -
  • clarity: -
  • formatting: -
  • grammar: -

Author:  Chengxiang Ding  on 2023-05-02  [id 3638]

(in reply to Report 2 on 2023-04-06)

Thanks very much!

We add a short paragraph to clarify this question (two lines above Eq. (9)):

It should be noted that the finite-size scaling corrections arise from two sources, one is the leading correction proportional to $L^{-\omega_1}$, with $\omega_1 = 0.759$ for the current model, which comes from the irrelevant scaling field, another one is the background contribution analytic in $L^{-1}$. In practice, the analytic term $L^{-1}$ cannot be distinguished from $L^{-\omega_1}$ in the fitting procedure due to their close exponents. We also tried the fitting with $\omega_1=1$, the fitting quality is slightly worse and the difference of the results is very small, which falls in the range of the uncertainty of the error bars. Such a strategy has also been applied to all the other data fittings in this and the next subsections, although not explicitly stated.

---

## Round 2 · Referee Report · Anonymous · 2023-4-8

Report

I thank the authors for their response to my comments. However, I still have several questions that I would like the authors to address:

1. In the definition of the Binder ratio in Eq. (7), should the numerator and denominator contain second power of `\tilde S`, instead of fourth power, since `\tilde S` already contains two powers of the order parameter? If so, is this a typo and is fig 2b) generated using the correct formula?

2. In an ordered phase, the Binder ratio should approach `1` as `L \to \infty`. Instead, in fig 2b) the Binder ratio at large `J_s` seems to be above `1.2` and, if anything, slightly increasing with the system size. Is there some explanation for this?

3. I think it would be useful to have a plot of `C_s(r)` in the paper (say, for the largest system size available). Perhaps this would shed some light on the disparity of `C_{||}(L/2)` and `{S(\pi)}/L` in Fig. 7.

4. I am confused by the discussion of `(\xi/L)^2` in the extraordinary phase at the end of section 3. In particular, I don't understand where the statement that `(\xi/L)^2` should scale as `L` for an ordered boundary is coming from. It is true that for a bulk quantum antiferromagnet in `d>1 ` spatial dimensions, `(\xi/L)^2 \propto L^{d-1}`, so d = 2 gives `(\xi/L)^2 \propto L`. But here the boundary is one-dimensional, so one might instead expect `(\xi/L)^2 =O(1)` (perhaps up to logarithmic corrections, as in the extraordinary-log phase).

Here is another comment about the analysis of `(\xi/L)^2`. The authors claim that there is a finite ordered moment in the extraordinary phase. Let me accept this claim for now. However, looking at Fig. 7, `{S(\pi)}/L` varies by some factor of `4` over the range of `L` studied, further the extrapolated value of the ordered moment is an additional factor of `3` below the value of `{S(\pi)}/L` at the largest `L` studied. So, if there is an ordered moment, the behavior of `{S(\pi)}/L` at system sizes studied is very far from the asymptotic, saturated behavior. Thus, it would be meaningless to compare `(\xi/L)^2` to predictions for asymptotic behavior in the ordered phase.

In addition, it seems that `{S(\pi)}/{S(\pi+\delta q)}` is numerically not very large for `L ` values studied (perhaps O(1)). Typically, the analysis of `\xi/L` in the ordered phase is performed assuming that this ratio is large, so that the `-1` term under the square root in Eq. (6) can be neglected.

For all these reasons, I would encourage the authors to show a plot of `S(\pi + \delta q)` as a function of `L` in the extraordinary phase by itself, without the additional algebraic manipulations that go into `\xi/L`.

  • validity: -
  • significance: -
  • originality: -
  • clarity: -
  • formatting: -
  • grammar: -

Author:  Chengxiang Ding  on 2023-05-02  [id 3639]

(in reply to Report 3 on 2023-04-08)

Thanks very much!

To question 1: It is a typo, we have fixed it.

To question 2:

We cannot thoroughly understand this result, and we think that the possible reasons may include but not limited to:

1. The surface is coupled with the bulk critical state, this is equivalent to effective long-range interactions for the surface, which may also have an impact on the value of the Binder Ratio.
In fact, in recent days, we have studied a long-range quantum Heisenberg chain by Monte Carlo simulations, it is shown the Binder Ratio in the long-range ratio is obviously larger than 1, the results will be published elsewhere.

2. Although the extraordinary phase is ordered, the value of the order parameter is very small; because of the existence of the gapless mode, the fluctuation of the AF order is still large, which makes it look like a critical state, hence the value of Binder ratio is not equal to 1.

We leave this question in future study.

To question 3:

It is a pity that we have only computed $C_\parallel(L/2)$.

P.S. We rewrite the definition of $C_\parallel$ as
$C_{\parallel}(r)=\frac{(-1)^r}{L}\sum_{x}\langle \bf{S}_{(x,1)}\cdot\bf{S}_{(x+r,1)}\rangle$ in Eq. (2); and all the $C_s$ in the text have been changed to $C_\parallel$.

To question 4:
We revisited the scaling behaviors of $S(\pi)$, $S(\pi+\delta q)$, and $(\xi_s/L)^2$ in the extraordinary phase (ordered), the paragraph after 7 lines of Eq. (27) is revised as:

Furthermore, according to the definition of Eq. (5), in an ordered phase, $S(\pi+\delta q)$ should growth logarithmically (the constant term cancels out after summing, and the integral of the $1/r$ term contribute the logarithmic term), i.e., the data of $S(\pi+\delta q)$ should satisfy the finite-size scaling form
\begin{eqnarray}
S(\pi+\delta q)=a+b\log(L);
\end{eqnarray}
combining with scaling of $S(\pi)$ in Eq. (26), we get the scaling formula of the square of the correlation ratio, which is written as
\begin{eqnarray}
(\xi_s/L)^2=a+bL/\log (L).
\end{eqnarray}
Figure 10 shows the scaling behaviors of $S(\pi+\delta q)$ and $(\xi_s/L)^2$ in the extraordinary phase, with $J_s=16$, which further demonstrates that there is a long-range AF order.

---

## Round 2 · Author Response

resubmitted version.

---

## Round 3 · List of Changes

1. We fixed a typo: should growth -> should grow.

2. We further revise the discussion part according to the suggestion of Referee-2, i.e.,

we replace:

“In Ref. [17], the extraordinary-log phase was proposed based on the perturbative RG analysis near the normal fixed point at the 1D boundary. Starting from the normal fixed point, where the spins show an infinitesimal long-range order, the spin interactions would be relevant and lead to short-range correlations at a free-standing boundary, but the coupling with the bulk critical modes reverses the RG flow direction and makes the normal fixed point stable.”

by

“In Ref. [17], the extraordinary-log phase was proposed based on the perturbative RG analysis near the ordered fixed point at the 1D boundary. Starting from the ordered fixed point, spin fluctuations would lead to short-range correlations for a free-standing boundary, but the coupling with the bulk critical modes reverses the RG flow direction and makes the ordered fixed point stable.”

3. As to the question of the citation of Ref. 22: As pointed out by the referee-2, this reference does not exactly prove the existence of the boundary of the AF order; however, in the RG analysis, it does show the possibility of such boundary AF order; therefore, we added words like “possible”, “possibility”, or “may” to somewhere the place Ref. 22 is cited to keep the rigor of the statement.

---

## Editorial Decision

published